



# Carbon dioxide fluxes and carbon balance of an agricultural grassland in southern Finland

Laura Heimsch[1], Annalea Lohila[1,2], Juha-Pekka Tuovinen[1], Henriikka Vekuri[1], Jussi Heinonsalo[1,3], Olli Nevalainen[1], Mika Korkiakoski[1], Jari Liski[1], Tuomas Laurila[1], and Liisa Kulmala[1,3]

[1]Finnish Meteorological Institute, P.O. Box 503, 00101 Helsinki, Finland
[2]Institute for Atmospheric and Earth System Research, Physics, P.O. Box 64, 00014 University of Helsinki, Finland
[3]Institute for Atmospheric and Earth System Research, Forest Sciences, P.O. Box 27, 00014 University of Helsinki, Finland

**Correspondence:** Laura Heimsch (laura.heimsch@fmi.fi)

**Abstract.** A significant proportion of the global carbon emissions to the atmosphere originates from agriculture. Therefore, continuous long-term monitoring of $CO_2$ fluxes is essential to understand the carbon dynamics and balances of different agricultural sites. Here we present results from a new eddy covariance flux measurement site located in southern Finland. We measured $CO_2$ and $H_2O$ fluxes at this agricultural grassland site for two years from May 2018 to May 2020. Especially the

first summer experienced prolonged dry periods, which affected the $CO_2$ fluxes, and substantially larger fluxes were observed in the second summer. During the dry summer, leaf area index (LAI) was notably lower than in the second summer. Water use efficiency increased with LAI in a similar manner in both years, but photosynthetic capacity per leaf area was lower during the dry summer. The annual carbon balance was calculated based on the $CO_2$ fluxes and management measures, which included input of carbon as organic fertilisers and output as yield. The carbon balance of the field was $-50 \pm 68$ g C m$^{-2}$ yr$^{-1}$ and $-118$

$\pm 24$ g C m$^{-2}$ yr$^{-1}$ in the first and second study year, respectively. We estimated that on average the grassland exceeded the global "4 per 1000" goal to increase the soil carbon content.

## 1 Introduction

Conventional and intensive agricultural practices cause significant carbon emissions while diminishing the soil organic matter (SOM) content. This leads to a reduction of soil quality and health (e.g. Houghton and Nassikas, 2017; Le Quéré et al., 2009,

2017; Lal, 2016; Paustian et al., 2000; Smith, 2008). Currently, agriculture is responsible for more than 10% of the global anthropogenic greenhouse gas (GHG) emissions to the atmosphere (Le Quéré et al., 2017). Soil type and properties, vegetation, climate and weather conditions as well as management practices all have a considerable effect on the carbon fluxes and balances of agroecosystems (Bolinder et al., 2010; Gomez-Casanovas et al., 2012; Jensen et al., 2017; Lorenz and Lal, 2018; Singh et al., 2018). Frequent ploughing, monocropping and intensive use of agrochemicals are the main contributors to the loss of SOM

and the resulting carbon dioxide ($CO_2$) emissions from land use (Ceschia et al., 2010; Reinsch et al., 2018; Yang et al., 2019). A change from conventional and intensive agricultural practices to regenerative and holistic farm management provides a substantial climate change mitigation potential (Lal, 2016). Increasing the amount of SOM in agroecosystems by applying enhanced management practices, such as lighter tillage, continuous plant cover, rotational grazing, agroforestry, increased





biodiversity and cover cropping, would not only help to mitigate climate change but also to restore soil quality and fertility.
Especially, managed grasslands as part of agricultural systems have a high potential for substantial soil carbon sequestration
(Soussana et al., 2010; Gilmanov et al., 2010; Yang et al., 2019). The importance of increasing soil organic carbon (SOC)
content of agricultural soils has recently attained more attention, and the "4 per mille Soils for Food Security and Climate"
initiative was launched at the 21$^{st}$ Conference of the Parties to the United Nations Framework Convention on Climate Change
in Paris in 2015 (Minasny et al., 2017). The aim of this initiative is to increase the soil carbon stock on all land surfaces in the
upper 2 metres by 0.4% annually. This would be enough to sequester carbon from the atmosphere by an amount equivalent to
the annual anthropogenic GHG emissions. However, the initiative states that the most potential SOC increases can be achieved
on managed agricultural lands. In that case, the "4 per 1000" means increasing of SOC at the top 1-m layer of agricultural soils
by 0.4% annually. That would effectively offset approximately 20–35% of the global GHG emissions.

Agricultural ecosystems are highly prone to impacts of climate change, which induces a risk for food production. One of the
possible impacts of climate change on agricultural ecosystems is associated with the changes in seasonal weather conditions and
the resulting alteration in the carbon and water balance of these ecosystems (Ciais et al., 2014; Donnelly et al., 2017; Harrison
et al., 2019). Severe drought events and storms causing considerable damage to agriculture have already been observed across
Europe (Ciais et al., 2005; Wolf et al., 2013; Bastos et al., 2020). Moreover, adverse climatic impacts may be amplified by
current and prior land use practices if they have not supported ecosystem resilience (Brunsell et al., 2014). For instance, a
deeper root system is likely to buffer the negative impacts of climate variability. Also, high plant species diversity, compared
to monocultures, favours the efficiency of plant water consumption and resilience to drought (De Boeck et al., 2006). As gross
primary production (GPP) is closely related to ecosystem evapotranspiration (ET) via stomatal functions (Fricker and Willmer,
2012), changes in terrestrial water balance are potentially reflected in GPP and thus in the carbon balance of agricultural
grasslands. The effect of water stress can be studied, for instance, by analysing ecosystem water use efficiency (WUE), i.e.
the amount of carbon assimilated per unit of water lost by ET (Steduto, 1996). Generally, the productivity of a grassland
ecosystem correlates with WUE, and thus ecosystems with a high productivity usually also have a high WUE (Hu et al., 2008).
Environmental factors are mainly regulating WUE via the effects on GPP, and during prolonged drought periods, for example,
temperature-induced downregulation of GPP may reduce WUE of grasslands (Gharun et al., 2020). Furthermore, the WUE
response depends on the intensity of the drought (Xu et al., 2019). However, the drought effects are also strongly related to
season as Wolf et al. (2013) reported that the WUE of Swiss grassland ecosystems did not respond to a spring drought and
Bastos et al. (2020) concluded that the spring weather may either amplify or dampen the carbon and water dynamics during
the following summer.

Better understanding of climatic impacts of agriculture and the effects of improved practices from the perspective of soil
health and vitality is needed in order to develop tools for better environmental management of these ecosystems. Continuous
long-term measurements of the atmosphere-ecosystem fluxes are needed to identify the key factors affecting carbon dynamics
of different ecosystems, to quantify the resulting carbon balance and its components, and to verify soil carbon and ecosystem
models. Moreover, the high-quality GHG flux data is needed for a reliable, global monitoring and verification system of
agricultural carbon fluxes and soil carbon sequestration and stability (Smith et al., 2020).





The eddy covariance (EC) method is widely used for measuring $CO_2$ and energy fluxes in different ecosystems and climatic conditions (Aubinet et al., 2012). The high-frequency measurements provided by EC allow a direct quantification and analysis of gas exchange between the ecosystem and atmosphere. The carbon balance calculated from EC data, combined with the additional carbon fluxes caused by management, serves as an important measure for determining the climatic impact of agricultural ecosystems (e.g. Baldocchi, 2003; Baldocchi et al., 2018). However, continuous GHG flux measurements on agricultural sites, especially on mineral soils and grasslands, are still scarce in the Nordic countries (Shurpali et al., 2009; Lind et al., 2020; Jensen et al., 2017).

The aim of this study is to determine the magnitude and seasonal dynamics of the carbon balance of a managed forage grassland in southern Finland. In particular, we had three specific research questions:

1. How does the $CO_2$ exchange and carbon balances vary between the study years?

2. Does the grass photosynthesis indicate occasional drought-related responses?

3. How does the possible carbon sink relate to carbon sequestration objective of "4 per 1000" initiative?

For the purposes of this study, we collected field data on the net exchange of $CO_2$ and $H_2O$, soil and vegetation properties and meteorological variables on an agricultural grassland in southern Finland during two years, from May 2018 to May 2020.

## 2 Material and methods

### 2.1 Site description

The flux measurements were conducted at the Qvidja farm in southern Finland (60.29550°N, 22.39281°E; elevation 5 m) from May 2018 to May 2020 (Fig. 1). The site belongs to the hemiboreal climate zone. From 1981 to 2010, the mean annual air temperature and precipitation at the Kaarina Yltöinen weather station, located 13 km northeast of Qvidja, were 5.4°C and 679 mm, respectively (Pirinen et al., 2012). The experimental field in Qvidja has mineral soil (clay loam) and it covers 16.25 ha. It was cultivated as forage grassland during the study years. From 2008 to 2016, the field was managed intensively with conventional practices, and it was in annual crop rotation. In 2017, the field management practices were converted towards more sustainable and environmentally friendly farming by increasing the use of organic fertilisers and perennials, restricting the use of pesticides and increasing plant species biodiversity. The current grass was sown as an undergrown species with broad bean in spring 2017. The predominant grass species were timothy (*Phleum pratense*), meadow fescue (*Festuca pratensis*) and white clover (*Trifolium repens*).

Grass was harvested for silage for the first time on 12 June 2018. As the grass cover was fairly sparse later in the summer due to drought, repair seeding was done on 3 September 2018 to restore the drought-induced damage. The seed mixture included 35% of timothy, 30% of rye-grasses (*Lolium spp.*), 20% of common meadow-grass (*Poa pratensis*) and 15% of red fescue (*Festuca rubra*). Timothy, meadow fescue and clover remained as the predominant species also in 2019 and early 2020. On 21 August 2018, the grass was cut at approximately 15 cm, but the yield was left in the field. The second harvest of 2018 occurred





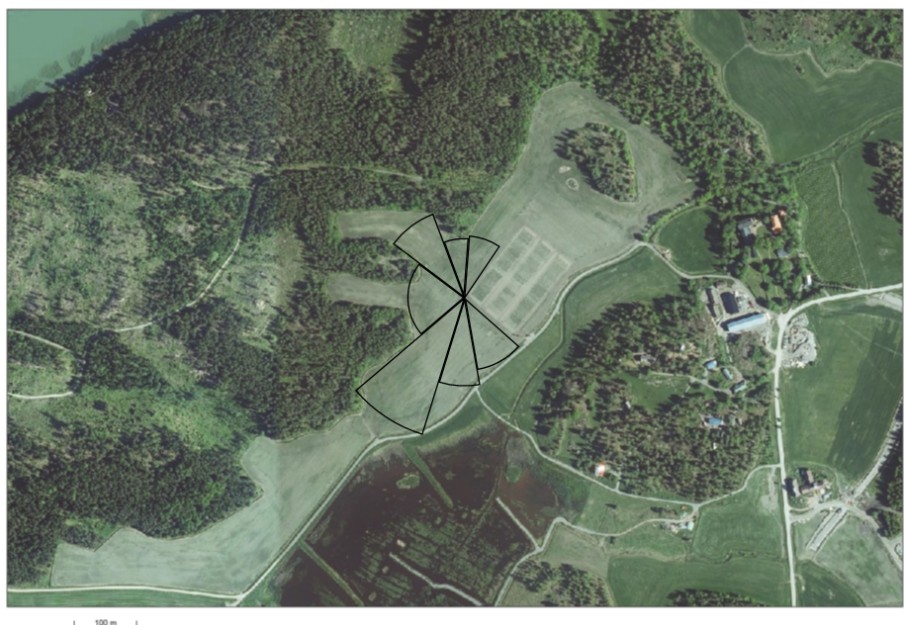

**Figure 1.** Experimental field with the sectors representing the target area. the area covers 3.9 ha. Eddy covariance tower is located in the centre of the sectors. Wind directions from 30 to 140° were filtered out due to another experimental plot locating in that part of the field. (Orthophoto from National Land Survey of Finland)

on 23 September. In 2019, the grass was harvested on 11 June and 20 August. In June 2018, a conventional cutting height of 6 cm was used, whereas in the other harvests the grass was cut at 15 cm.

In 2018, the field was fertilised twice, on 16 July and 24 August, with 2800 kg ha$^{-1}$ and 1800 kg ha$^{-1}$ of NK-molasses, respectively (Table 1). NK-molasses was a byproduct of the sugar industry. It contained 67% of organic matter (OM) and 4.4% of nitrogen and had the C:N ratio of 9. According to the product information, the molasses included 205 g kg$^{-1}$ of organic

carbon. In addition, it contained potassium and small proportions of sulphur, magnesium, calcium and sodium.

In May 2019, the field was fertilised with a mixture of side products from industries of starch potato processing, biowaste processing and ethanol production out of sawdust. This fertilisation mixture contained 65% of OM, 1.3% of nitrogen, 0.2% of phosphorus, 3% of potassium and 0.4% of sulphur, as well as small amounts of calcium, magnesium, zinc, copper and manganese. Approximately 4600 kg ha$^{-1}$ was applied on the field on 8 May (Table 1). On 26 June after the first harvest, 220

100  kg ha$^{-1}$ of mineral fertiliser was applied. This fertiliser contained 23% of nitrogen, 10% of phosphorus and 8% of potassium.

## 2.2  Measurement setup

The CO$_2$ and H$_2$O fluxes were measured with the micrometeorological EC method. The flux measurements started on 3 May 2018, and here we analysed data collected from 4 May 2018 to 3 May 2020. From this point on, the periods of 4 May 2018 – 3 May 2019 and 4 May 2019 – 3 May 2020 are referred to as the first and second EC measurement year, respectively.





The EC instrumentation consisted of an enclosed infrared $CO_2$/$H_2O$ gas analyser (LI-7200, LI-COR Biosciences, NE, USA), which detects the $CO_2$ and $H_2O$ mixing ratios, and a three-dimensional sonic anemometer (uSonic-3 Scientific, METEK GmbH, Elmshorn, Germany) to measure wind speed and air temperature. The data were recorded at 10-Hz frequency. The measurement height was 2.3 m. The flow rate was about $12\,l\,min^{-1}$, and the length of the 4-mm stainless steel inlet tube with $2\,\mu m$ Swagelok sinter was 0.8 m. The gas analyser was calibrated with a zero $CO_2$ concentration air as a reference gas in May 2018 and March 2020. The micrometeorological sign convention is used throughout the paper, with a negative value indicating the flux from the atmosphere to the ecosystem (net uptake) and a positive value indicating the flux from the ecosystem to the atmosphere (net emission).

Auxiliary meteorological measurements were conducted next to the flux tower. These included soil moisture observations at the depth of 0.1 m (ML3 ThetaProbe sensor, Delta-T Devices Ltd., Cambridge, UK) and soil temperature profile at the depths of 5, 10 and 30 cm (Pt100 IKES sensors, Nokeval Oy, Nokia, Finland). The soil temperature data were collected with a Vaisala QML201C datalogger (Vaisala Oyj, Vantaa, Finland). Photosynthetically active radiation (PQS PAR sensor, Kipp & Zonen B.V., Delft, The Netherlands), global and reflected solar radiation (CMP3 radiometer, Kipp & Zonen) and air temperature (Humicap HMP155, Vaisala Oyj) were measured at the height of 1.8 m. In addition, precipitation was measured with Pluvio2 (OTT HydroMet GmbH, Kempten, Germany). Meteorological measurements started on 8 May 2018, and the data were recorded as 30-min averages, excluding the precipitation which was recorded as 1-min values. Snow cover was recorded at the weather station of Kaarina Yltöinen.

The leaf area index (LAI) data were obtained from the Sentinel-2 satellite as daily values on the clear-sky days. LAI was calculated from the Sentinel-2 bottom-of-atmosphere products (L2A) using the Google Earth Engine (GEE) and a Python implementation of the Biophysical Processor toolbox (Weiss and Baret, 2016) available in Sentinel Application Platform (SNAP) software. The cloudy, cloud-shadowed and snowy data were filtered out using the scene classification band available in the L2A products.

## 2.3 Eddy covariance data processing

The turbulent fluxes were determined as the covariance between the variations of vertical wind component and gas mixing ratio recorded at 10 Hz. They were calculated as 30-min block averages applying standard procedures, including double coordinate rotation and lag determination based on cross-correlation analysis (Rebmann et al., 2012). The systematic flux loss due to the incomplete frequency response of the measurement system was corrected according to the empirical method described by Laurila et al. (2005).

The EC data from 5 January to 28 March 2019 were affected by technical issues with an inlet filter, which resulted in an erroneous reading of the internal analyser pressure. For this period, the 10-Hz mixing ratios were recalculated from the recorded absorptance data using the instrument-specific calibration functions. The mean $CO_2$ mixing ratio was set to 410 ppm in these calculations. The following acceptance criteria were applied to screen the 30-min averaged $CO_2$ flux data: number of spikes in the raw data < 150 of 18,000, relative stationarity of $CO_2$ flux (Foken et al., 2012) < 50%, mean $CO_2$ mixing ratio > 380 ppm, variance of $CO_2$ mixing ratio < 15 $ppm^2$ between April and September and < 5 $ppm^2$ between October and March, and wind





direction within 0–30° or 140–360°. Furthermore, the data were discarded during the periods of weak turbulence and when the
flux footprint was not sufficiently representative of the target grassland, as estimated with the footprint model of Kormann and
Meixner (2001). For these, we applied a friction velocity limit of 0.06 m s$^{-1}$ and a cumulative footprint limit of 0.7. The further
screening applied to $H_2O$ fluxes included: $H_2O$ flux > 0, relative stationarity of $H_2O$ flux < 50% and variance of $H_2O$ mixing
ratio < 1 (mmol mol$^{-1}$)$^2$. After applying these filtering criteria, the coverage of $CO_2$ and $H_2O$ flux data accepted for further
analysis was 44% and 30% of all the 30-min periods during the two measurement years, respectively. Most of the accepted
$CO_2$ and $H_2O$ flux data were collected when the wind direction was in the south-southwest sector (Fig. 2).

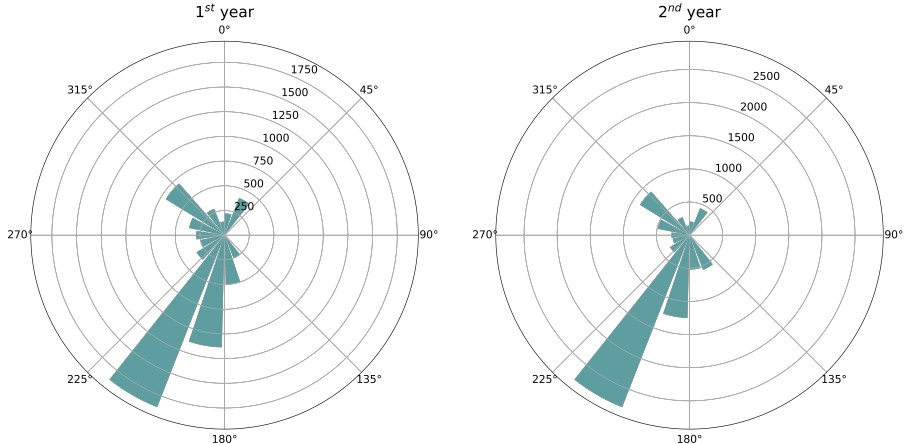

**Figure 2.** Number of accepted flux measurements within 20° sectors around the flux tower during the first and second year. Data from 30°
to 140° were discarded.

## 2.4 Soil temperature model

The soil temperature sensor at the depth of 5 cm malfunctioned during the first measurement year, and these data were re-
placed with values derived from air temperature using the model presented by Rankinen et al. (2004). This model also takes
into account the effect of possible snow cover on soil temperature. The following equation was used to obtain 30-min soil
temperatures at 5 cm from 8 May 2018 to 3 May 2019:

$$T_z^{t+1} = T_z^t + \left[ \frac{\Delta t K_T}{C_A(2Z_s)^2}(T_{air}^t - T_z^t) \right] e^{-f_s D_s} \tag{1}$$

where $T_z^{t+1}$ is the soil temperature at the depth of $Z_s$ on the following day, $T_z^t$ is the soil temperature of the current day, $\Delta t$ is
the length of the timestep, $K_T$ is soil thermal conductivity, which was set to 1 W m$^{-1}$ K$^{-1}$, $C_A$ is the apparent heat capacity
which is the sum of specific heat capacity of the soil $C_s = 0.5 \times 10^{-6}$ J m$^{-3}$ K$^{-1}$ and specific heat capacity due to freezing
and thawing $C_{ice} = 4 \times 10^{-6}$ J m$^{-3}$ K$^{-1}$, and $T_{air}^t$ is the measured air temperature. The impact of snow cover was taken into
account in the last term of the equation where $f_s$ is an empirical snow parameter, which was set to 10 m$^{-1}$, and $D_s$ is the





measured snow depth. The model predictions were compared to measurements at the experimental field between June 2019 and May 2020. During summertime, the changes in soil temperature were fairly well captured by the model, whereas in the wintertime, the model tended to create larger changes in temperature than the actual measurements showed (Fig. 3).

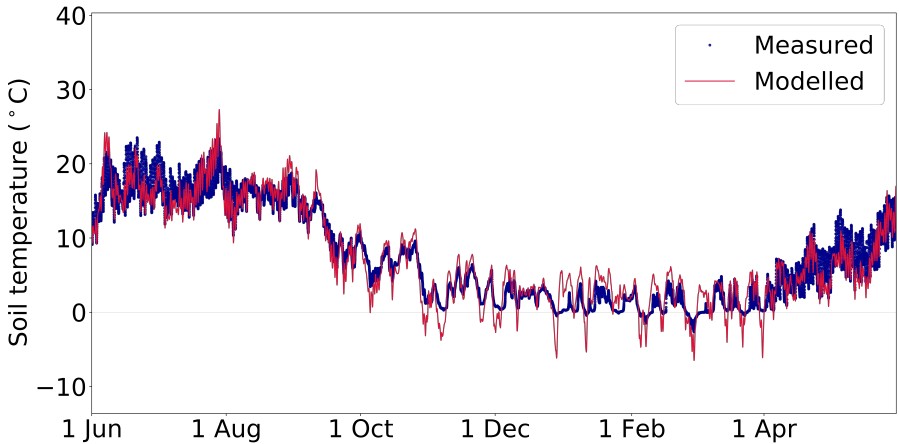

**Figure 3.** Comparison of measured and modelled soil temperature at 5 cm depth from June 2019 to May 2020.

## 2.5 Flux partitioning and gap-filling

To calculate $CO_2$ balances and to conduct further seasonal analysis of the flux components the measured $CO_2$ flux data (i.e. net ecosystem exchange, NEE) were partitioned to GPP and total ecosystem respiration ($R_{eco}$) and gap-filled based on this partitioning:

$$NEE = GPP + R_{eco} \qquad (2)$$

The gap-filled GPP and $R_{eco}$ were calculated with empirical response functions by first fitting these functions to the flux data. $R_{eco}$ was expressed as a function of temperature (Lloyd and Taylor, 1994):

$$R_{eco} = R_0 e^{E_0 \left( \frac{1}{T_1} - \frac{1}{T_s - T_0} \right)} \qquad (3)$$

where $R_0$ is the respiration rate (mg m$^{-2}$ s$^{-1}$) at the reference soil temperature of 283.15 K, $T_0 = 227.13$ K, $T_1 = 56.02$ K, and $E_0 = 308$ K is the long-term ecosystem sensitivity coefficient (Lloyd and Taylor, 1994) that describes the temperature response
of soil respiration, and $T_s$ is the soil temperature at the depth of 5 cm.



GPP was modelled as a function of photosynthetically active radiation (PAR, $\mu$mol m$^{-2}$ s$^{-1}$) and daily effective phytomass index (PI) as:

$$GPP = PI \times \frac{\alpha \times PAR \times GP_{max}}{\alpha \times PAR + GP_{max}} \tag{4}$$

where *PI* is an empirically-determined variable introduced to describe the seasonal changes in the photosynthetically active vegetation (Aurela et al., 2001), $\alpha$ is the apparent quantum yield (mg $\mu$mol$^{-1}$), and $GP_{max}$ denotes the asymptotic CO$_2$ uptake rate in optimal light conditions (mg m$^{-2}$ s$^{-1}$). Further details on the PI determination and gap-filling procedure are provided in the Appendix A and B, respectively. Energy fluxes were gap-filled following the description in the Appendix C.

To study the differences in photosynthetic capacity of the grass field between the two growing seasons, daily GP$_{1200}$ values
were calculated with the estimated $\alpha$ and GP$_{max}$ values, i.e. GPP was normalised to PAR = 1200 $\mu$mol m$^{-2}$ s$^{-1}$.

### 2.6 Carbon balance

The carbon balance of the agricultural ecosystem was calculated by adding up the 30-min NEE fluxes, the imported carbon in the form of organic fertilisers and the removal of carbon as harvested biomass:

$$C_{balance} = C_H + C_F + \sum_{i=1}^{m} NEE_i \tag{5}$$

where $C_H$ is the amount of carbon in harvested biomass, $C_F$ is the amount of carbon in imported fertilisation and *m* is the total number of timesteps in the period for which the balance was calculated. Biomass was converted to carbon by multiplying the dry weight by 0.42 (Lohila et al., 2004). The following sign convention was used: the carbon imported into the ecosystem corresponds to a negative flux and the carbon removed from the system corresponds to a positive flux.

### 2.7 Uncertainty analysis

The CO$_2$ balance, which is calculated based on the EC measurements, includes multiple potential error sources. Uncertainties are associated, for example, with the stochastic nature of turbulence and incomplete sampling of large eddies, the performance of instruments and the flux variation caused by the limited area of the target ecosystem (Aubinet et al., 2012). Some of these errors were compensated for in the data processing and screening. Here we included in the uncertainty estimate the most



relevant random error sources, i.e. the statistical measurement error ($E_{meas}$) and the error caused by gap-filling ($E_{gap}$) Aurela
et al. (2002):

$$E_{meas} = \sqrt{\sum_{i=1}^{n} (NEE_{meas,i} - NEE_{mod,i})^2} \qquad (6)$$

where $NEE_{meas}$ is the filtered 30-min flux, $NEE_{mod}$ is the corresponding modelled NEE (Eqs. 2–4), and $n$ is the number of
measured data.

$$E_{gap} = \sqrt{\sum_{i=1}^{N} (E_{GPP,i}^2 + E_{R_{eco},i}^2)} \qquad (7)$$

where $E_{GPP}$ and $E_{R_{eco}}$ are the errors of modelled GPP and $R_{eco}$, respectively. $N$ is the number of gaps in the data.

The standard error propagation principle was used in estimating the total uncertainty ($E_{tot}$) of the annual carbon balance:

$$E_{tot} = \sqrt{E_{meas}^2 + E_{gap}^2} \qquad (8)$$

## 2.8 Water use efficiency

The ecosystem WUE was defined as the ratio of GPP to ET, i.e. $H_2O$ flux:

$$WUE = \frac{GPP}{ET} \qquad (9)$$

where daily means of GPP and ET were used. The ET data corresponding to a latent heat flux lower than 30 W m$^{-2}$ were
discarded (Abraha et al., 2016).

## 2.9 Soil carbon content

Soil carbon content was determined from 1-m-deep core samples taken within the flux source area. The samples were taken
using a hydraulic corer installed to a tractor in October 2018. The diameter of the sample cylinder was 151 mm. Subsamples





were taken along the 1-m core at 16 points, and soil organic carbon (SOC, kg m$^{-2}$) content in each subsample was analysed
using a VarioMax CN analyser (Elementar Analysensysteme GmbH, Germany).

## 3 Results

### 3.1 Meteorological conditions

The annual mean air temperature at the study site was 7.6 °C and 7.7 °C in the first and second measurement year, respectively.
Both years were warm compared to the long-term (1981–2010) average of 5.4 °C measured at a nearby weather station (Pirinen
et al., 2012). The annual precipitation sum was lower in the first year (473 mm) and higher in the second year (855 mm) than
the long-time average (679 mm).

The thermal growing season, defined here as the period when the daily mean temperature exceeded permanently 5 °C, started
on 14 April in 2018, i.e. before the EC measurements started. In 2019 and 2020, the thermal growing season began on 16 April
and 18 April, respectively. The thermal growing season ended on 17 November and 26 October in 2018 and 2019, respectively.
Thus, the thermal growing season length was 218 days in 2018 and 194 days in 2019. Meteorological conditions during the
main growing season between May and September varied substantially between the two years. The mean air temperature
during these months was 16.7 °C and 14.5 °C in 2018 and 2019, respectively. The mean growing season soil temperatures
were similar to the air temperatures with 16.4 °C (modelled) in 2018 and 14.5 °C in 2019. During the same period, the mean
daily PAR was about 12% higher in 2018 than in 2019 (460 vs. 410 $\mu$mol m$^{-2}$ s$^{-1}$), while the precipitation sum was 32%
lower (212 vs. 312 mm).

During winter 2018–2019, permanent snow cover was recorded from 17 December to 26 March 2019. In the following
winter (2019–2020), there were only two short snow-cover periods: 5–8 February and 30–31 March 2020. The maximum
snow depth in the first winter was 33 cm, whereas in the second winter it was 3 cm. The mean wintertime (November–March)
air temperature was –0.2°C in 2018–2019 and 2.2 °C in 2019–2020. The warmer winter in the second measurement year was
also observed in the mean soil temperature (–0.9 vs. 1.6 °C).

Soil moisture content at the depth of 10 cm varied between 0.16 and 0.55 m$^3$ m$^{-3}$ during the study period. In several
occasions, the daily mean soil moisture dropped to about 0.2 m$^3$ m$^{-3}$. During the growing seasons, such low values indicate
substantial drought, while in the winter, rapid data drops were likely related to soil freezing. The average soil moisture during
the growing season in 2019 was higher than in 2018 (0.30 vs. 0.26 m$^3$ m$^{-3}$). As a result of the higher precipitation in 2019,
soil moisture occasionally increased up to 0.4 m$^3$ m$^{-3}$, i.e. close to the saturated values observed in winter.

### 3.2 Fluxes

At the beginning of the measurements, the net $CO_2$ fluxes were negative (Fig. 4), and the air and soil temperatures were already
well above 10°C (Fig. 5). Net uptake was observed until the first harvest around mid-June 2018. This harvest and the following
management events during the that growing season induced large short-term variations in the $CO_2$ fluxes. Similarly, in the





second study year, large impacts on $CO_2$ fluxes were observed after the management events. During the growing season, the mean NEE was –0.13 and –0.21 mg $CO_2$ m$^{-2}$ s$^{-1}$ in 2018 and 2019, respectively. During the wintertime, no significant $CO_2$ uptake occurred, and the positive fluxes were small compared to the nocturnal fluxes in summer. The mean measured NEE between December 2018 and February 2019 was 0.03 mg $CO_2$ m$^{-2}$ s$^{-1}$, and during the same period in 2019–2020 it was 0.04 mg $CO_2$ m$^{-2}$ s$^{-1}$.

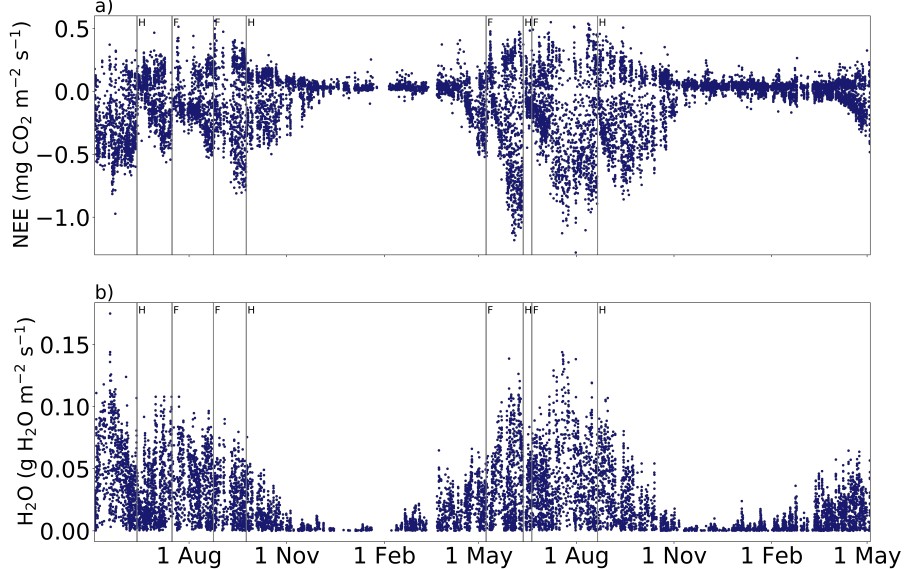

**Figure 4.** Accepted 30-min a) net ecosystem exchange (NEE) and b) $H_2O$ flux measurements from May 2018 to May 2020. Vertical lines with H and F indicate harvest and fertilisation, respectively.

Seasonal patterns were observed also in the $H_2O$ fluxes (Fig. 4). In the spring, the ecosystem ET started to increase reaching the highest levels between June and August, after which it gradually decreased to wintertime values, i.e. close to zero. The mean growing season $H_2O$ flux was 34.7 mg $H_2O$ m$^{-2}$ s$^{-1}$ in 2018 and 35.5 mg $H_2O$ m$^{-2}$ s$^{-1}$ in 2019. The wintertime (December– February) mean $H_2O$ flux was 3.6 mg $H_2O$ m$^{-2}$ and 3.7 mg $H_2O$ m$^{-2}$ in 2018–2019 and 2019–2020, respectively.

    The experimental field was harvested and fertilised twice during each of the studied growing seasons (Table 1). The effect of
management was investigated by comparing the mean fluxes 5 days before and after the harvest dates (Table A1). The harvest in June 2018 changed the mean $CO_2$ flux from a net sink of –0.28 mg $CO_2$ m$^{-2}$ s$^{-1}$ to a source of 0.03 mg $CO_2$ m$^{-2}$ s$^{-1}$, i.e. increased the net efflux by 0.31 mg $CO_2$ m$^{-2}$ s$^{-1}$. The first harvest of 2019 increased NEE by 0.47 mg $CO_2$ m$^{-2}$ s$^{-1}$, but as the pre-harvest mean NEE was –0.50 mg $CO_2$ m$^{-2}$ s$^{-1}$, the field remained as a net sink. As a result of the second harvest on 23 September 2018, the mean sink reduced from –0.10 to –0.02 mg $CO_2$ m$^{-2}$ s$^{-1}$, while the harvest on 20 August 2019
caused the sink to change from –0.25 to –0.02 mg $CO_2$ m$^{-2}$ s$^{-1}$. Thus, after all the harvests with a cutting height of 15 cm, the mean sink rate was diminished to –0.02 or –0.03 mg $CO_2$ m$^{-2}$ s$^{-1}$.



In the first growing season, the first and second fertilisation events with organic substances increased NEE by 0.27 and 0.08 mg $CO_2$ m$^{-2}$ s$^{-1}$, respectively, i.e. diminished the $CO_2$ sink (Fig. 4, Table A1). During the 5 days after the harvest in May 2019, the field acted as a $CO_2$ source. A similar trend was not observed in June 2019, as mineral fertiliser was used and thus no organic substances were added to the soil. Each of the fertilisation events were followed by rain within the next 5 days. However, the mean soil moisture remained either the same or decreased slightly (Fig. 5, Table A1). Furthermore, the mean air temperature increased after the fertilisations in July 2018 and May 2019, potentially affecting $CO_2$ fluxes. After the fertilisation events with organic substances in July 2018, August 2018 and May 2019, the mean PAR was 7%, 29% and 12% lower, respectively, than the 5-day mean before the fertilisation, complicating the interpretation of fertilisation impacts on the $CO_2$ fluxes. The effect of management on $H_2O$ fluxes could not be disentangled from the present data (Fig. 4b).

The PI calculated from the flux data was consistent with the seasonal changes in the LAI derived from Sentinel-2 images (Fig. 5d). The higher LAI in 2019 indicated that there was more photosynthesising green biomass before the first and second harvest compared to 2018. The effect of larger leaf area was also observed in the differences in the photosynthetic capacity (GP$_{1200}$) of the grassland between the study years (Fig. 6a). The years differed significantly ($p < 0.05$) in terms of GP$_{1200}$ at all levels of LAI (>1). Larger LAI values were observed throughout 2019, indicating that grass was growing better than in 2018. Furthermore, the grassland was photosynthesising more efficiently with the same leaf area in 2019 than in the previous year (Fig. 6a).

### 3.3 Water use efficiency

The ecosystem WUE estimate showed different seasonal variation during the studied growing seasons (Fig. 7). Generally, WUE was higher in 2019 than in 2018 throughout the growing season. WUE increased before the first harvest around mid-June in both years, indicating more efficient $CO_2$ uptake in terms of water use than during the spring. The 5-day mean WUE before the first harvest was 2.6 and 2.9 g $CO_2$ (kg $H_2O$)$^{-1}$ in 2018 and 2019, respectively. Due to the harvest, it dropped to 0.8 g $CO_2$ (kg $H_2O$)$^{-1}$ in 2018 and to 2.2 g $CO_2$ (kg $H_2O$)$^{-1}$ in 2019. During the latter growing season, WUE increased steadily towards 4 g $CO_2$ (kg $H_2O$)$^{-1}$ until the second harvest in August, whereas WUE remained predominantly below 2 g $CO_2$ (kg $H_2O$)$^{-1}$ during the same period in 2018. In the end of August and early September, WUE was at the same level in both years.

The LAI derived from the Sentinel-2 data was compared to the daily WUE values (Fig. 6b) to further interpret the relation between vegetation status and ecosystem WUE. While WUE was on average lower in 2018 than 2019, the difference at a given LAI was not significant ($p > 0.05$). However, in both years the daily WUE increased in a similarly linear manner in relation to LAI.

### 3.4 Carbon balance and soil carbon content

The carbon balance of the studied grass field was –50 $\pm$ 68 g C m$^{-2}$ yr$^{-1}$, i.e. not different from zero, in the first year, while the balance of the second year was negative, –118 $\pm$ 24 g C m$^{-2}$ yr$^{-1}$, i.e. the field acted as a net carbon sink (Table 2). All the components of the carbon balance were smaller in the first year than in the second one, GPP by 30%, R$_{eco}$ by 25% and management by 94%.





**Figure 5.** Daily mean a) air and soil (depth = 0.05 m) temperature, b) photosynthetically active radiation (PAR), c) precipitation and soil moisture (depth = 0.1 m), d) phytomass (PI) and leaf area indices (LAI), and e) daily mean NEE, GPP, $R_{eco}$ and cumulative carbon flux from May 2018 to May 2020.





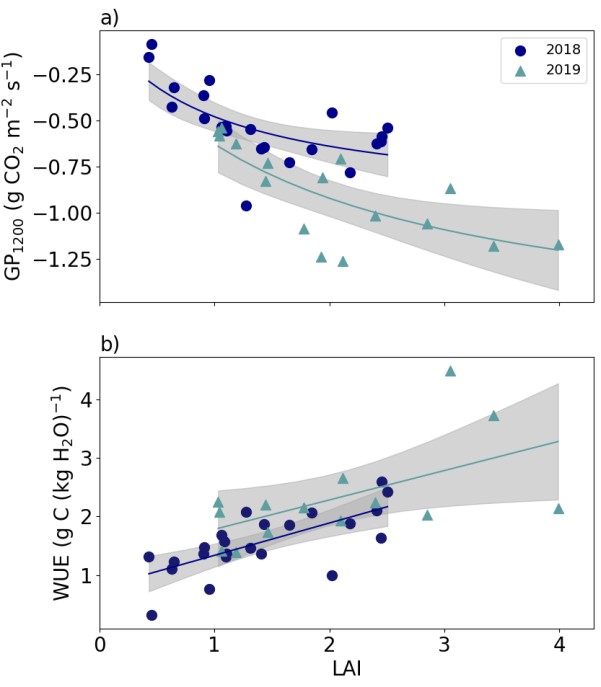

**Figure 6.** a) Daily photosynthetic capacity ($GP_{1200}$) and b) water use efficiency (WUE) as a function of leaf area index (LAI) during the two growing seasons. Grey areas represent the uncertainty estimation.

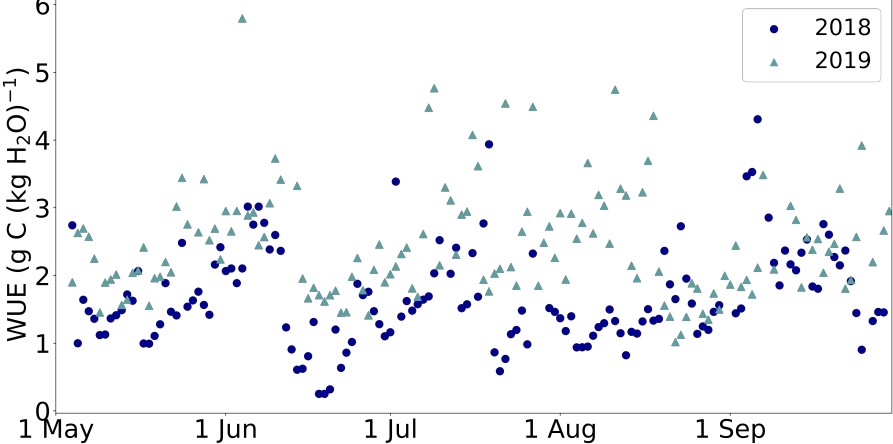

**Figure 7.** Daily water use efficiency (WUE) during two growing seasons.





**Table 1.** Different management events and their C inputs (fertilisation) and C outputs (harvest). During the cutting in August 2018, the grass was not collected and thus did not result to any C flux allocated to management.

| Date | Management | Output (dry weight kg ha$^{-1}$) | Input (kg ha$^{-1}$) | Carbon (g m$^{-2}$) |
|---|---|---|---|---|
| 12 Jun 2018 | Harvest | 1985 | | 83 |
| 16 Jul 2018 | Fertilisation | | −2800 | −57 |
| 21 Aug 2018 | Cutting | – | – | – |
| 24 Aug 2018 | Fertilisation | | −1755 | −36 |
| 23 Sep 2018 | Harvest | 348 | | 15 |
| 8 May 2019 | Fertilisation | | −4606 | −93 |
| 11 Jun 2019 | Harvest | 3107 | | 130 |
| 20 Jun 2019 | Fertilisation (mineral) | – | – | – |
| 20 Aug 2019 | Harvest | 1029 | | 43 |

**Table 2.** Annual carbon balances (g C m$^{-2}$ yr$^{-1}$) for the two measurement years. Negative values indicate C input into the ecosystem, whereas positive values indicate C loss. Management (M) is the sum of the C fluxes due to harvest (positive) and fertilisation (negative) events (Table 1). The values after ± represent the uncertainty in NEE.

| | NEE | GPP | R$_{eco}$ | M | Total balance |
|---|---|---|---|---|---|
| First year | −55 | −1034 | 972 | 5 | −50±68 |
| Second year | −198 | −1480 | 1291 | 80 | −118±24 |

There was a major difference in the CO$_2$ balances between the growing seasons (Table 3). In 2019, the growing season net uptake was 78%, GPP 49% and R$_{eco}$ 42% higher than in 2018.

The average soil carbon content in the 1-m layer was 16.59 ± 2.25 kg m$^{-2}$ (average ± standard deviation), with the highest SOC found in the top 30-cm layer (Fig. 8). The carbon balance of 2018 was 0.3% of the average SOC, and in 2019 this ratio was 0.7%. On average, the annual carbon input to the soil accounted for 0.5% of the SOC.

**Table 3.** Growing season (from 4 May to 30 September) CO$_2$ balances (g CO$_2$ m$^{-2}$) and total ET (mm) in 2018 and 2019.

| | NEE | GPP | R$_{eco}$ | ET |
|---|---|---|---|---|
| 2018 | −650 | −3190 | 2510 | 297 |
| 2019 | −1160 | −4740 | 3560 | 283 |





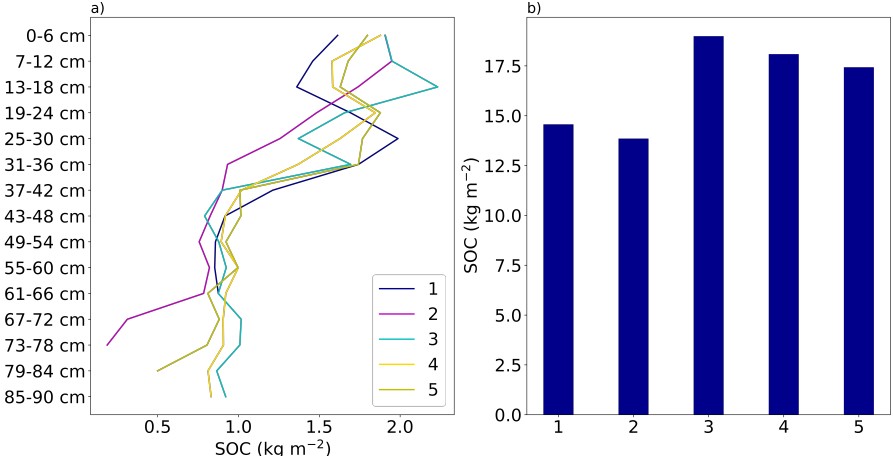

**Figure 8.** a) Soil organic carbon (SOC) content at different depths in the 1-m deep soil samples, and b) the total SOC in the samples. Numbers from 1 to 5 indicate sample numbers.

## 4 Discussion

### 4.1 Fluxes and carbon balance

The carbon fluxes on an agricultural grassland at Qvidja farm in southern Finland were clearly different between the two study years. The annual NEE was –55 g C m$^{-2}$ yr$^{-1}$ in the first study year (4 May 2018 – 3 May 2019) and –198 g C m$^{-2}$ yr$^{-1}$ in the second year (4 May 2019 – 3 May 2020). The GPP showed notable variation between the study years as the annual GPP was –1034 and –1480 g C m$^{-2}$ yr$^{-1}$ in the first and second year, respectively. Gilmanov et al. (2010) have reported the GPP of European managed grasslands to be in the range of –2107 to –1410 g C m$^{-2}$ yr$^{-1}$. Our results fall in to the lower range or below these GPP values. The annual R$_{eco}$ in Qvidja was also varying between the study years (972 and 1291 g C m$^{-2}$ yr$^{-1}$). Globally, the annual R$_{eco}$ of managed grasslands is reported to vary within a wide range from 31 to 2150 g C m$^{-2}$ yr$^{-1}$. The average R$_{eco}$ was 1445 and 647 g C m$^{-2}$ yr$^{-1}$ on the intensively and extensively managed grasslands, respectively (Gilmanov et al., 2010), and the annual R$_{eco}$ in Qvidja falls between these values in both study years. Regarding only the European grasslands, the annual R$_{eco}$ is reported to vary between 494 and 1623 g C m$^{-2}$ yr$^{-1}$ (Gilmanov et al., 2007). Our observations are thus also within this range.

The carbon balance of the grass field ecosystem in Qvidja was close to neutral (–50 ± 68 g C m$^{-2}$ yr$^{-1}$) in the first study year (4 May 2018 – 3 May 2019), and in the second year (4 May 2019 – 3 May 2020) the field was a moderate carbon sink (–118 ± 24 g C m$^{-2}$ yr$^{-1}$). Carbon balances, including the carbon equivalent of N$_2$O, CH$_4$ and management-related carbon fluxes, have been widely studied in other European agricultural grassland sites between 2002–2004 (Soussana et al., 2007). These nine grassland sites acted mainly as net carbon sinks, the annual net carbon balance ranging from –446 to 251 g CO$_2$–C eq. m$^{-2}$ yr$^{-1}$, where 13 of the 17 measured annual balances were negative. Our study site falls into this range. In comparison, the Finnish agricultural sites measured so far are generally carbon sources (Heikkinen et al., 2013; Shurpali et al.,





2009; Lind et al., 2016; Lohila et al., 2004). Lind et al. (2016) reported slightly higher annual net uptake of atmospheric $CO_2$ (two-year average NEE –259 g C m$^{-2}$ yr$^{-1}$) for a grassland site on mineral soil than we observed in Qvidja. However, by considering the total carbon balance of the system by taking into account the carbon fluxes caused by management, it was concluded that their site acted as a net carbon source. Mineral fertilisers were used during their study, and thereby no carbon was imported to the field to compensate for the biomass removal from the system as harvests. Similar carbon flux patterns

related to management were reported by Eichelmann et al. (2016). The annual NEE of the agricultural grassland in their study in Canada was more negative (average NEE –405 g C m$^{-2}$ yr$^{-1}$) than the NEE in Qvidja. However, the two-year mean annual carbon balance of the Canadian field was positive when biomass removal was taken into account, i.e. the field was a net source of carbon. It is noteworthy that the yield in Qvidja was substantially smaller than at the other two study sites (Lind et al., 2016; Eichelmann et al., 2016), at which the total balance became positive when the management was taken into account. However,

as the harvested grass was used as feed for farm animals, there was no need for a higher yield at Qvidja in either of the study years.

  Analysis of the weather variables in Qvidja indicated that temperature and moisture conditions were associated with the differences in fluxes and carbon balance between the study years. The growing season was warmer and drier in 2018 than 2019, with 13% lower mean soil moisture, 32% lower precipitation, 2.2 °C higher mean air temperature and 12% higher

radiation during the growing season, and substantially smaller fluxes were observed in the first year. This is in accordance with Shurpali et al. (2009) who observed a positive correlation between the uptake of $CO_2$ and both soil moisture and air temperature on another Finnish agricultural grassland. According to their conclusions, moderate temperature with high soil moisture favoured $CO_2$ uptake. Furthermore, Flanagan et al. (2002) and Kurc and Small (2007) concluded that photosynthesis of grassland favours rather wet summer conditions. These findings would support the conclusion that low soil moisture and

high temperatures were the main factors limiting $CO_2$ uptake at our study site in the summer 2018.

  To answer our first research question, we conclude that there were notable year-to-year differences in the carbon balances, but the reason behind this variation remains partly open, as weather, grass age and grass leaf area all showed different dynamics between the study years. In Finland, it is typical to grow grasslands for 3–4 years before the grass renewal. In Qvidja, the grass was not renewed between the study years, which may have led to the larger fluxes observed in the second year when the grass

root system, for instance, was possibly more developed enhancing carbon uptake. Furthermore, the leaf area was larger, and other capabilities, such as microbial symbioses (e.g. de Vries et al., 2020; Harman and Uphoff, 2019; Moreau et al., 2019), of the more developed grass may have increased carbon uptake. The lower leaf area during the first year was most probably also connected to the dry summer, as shortage of water is a growth-limiting factor. Besides the leaf area, the photosynthetic potential per leaf area was lower in the first year, indicating either drought stress or shortage of nutrients, as temperature, a

widely limiting factor in northern latitudes, was high enough during both summers not to restrict photosynthesis. In any case, a more specific analysis of the driving and inhibiting environmental factors will require a longer measurement period.

  Our second research question concerned the drought-related restrictions of photosynthesis. It has been widely recognised that in dry conditions plants are able to reduce transpiration by stomatal regulation (Pirasteh-Anosheh et al., 2016). However, grasses seem to limit stomatal functions only in severe, prolonged drought conditions (Wolf et al., 2013; Xu et al., 2019), and





thus occasional or seasonal drought events may not be observed in the ecosystem WUE of grasslands. In our study, WUE
values were predominantly lower in 2018 than in 2019. This was most probably explained by the differences in LAI as the
relationship between WUE and LAI was similar in both growing seasons (Fig. 6b). Furthermore, the drier conditions with
high temperatures in the summer 2018 may have resulted in a decoupling of assimilation and transpiration and in temperature-
induced downregulation of GPP (Gharun et al., 2020), as ET was similar in both years (Table 3). Therefore, the clearly lower

leaf-area-based photosynthetic capacity ($GP_{1200}$) in 2018 compared to 2019 probably indicates drought related stress in pho-
tosynthetic processes despite the similar leaf-area-based WUE (Fig. 6). It is noteworthy that the WUE analysis was performed
by means of the total ecosystem ET rather than plant transpiration, which would have enabled a more direct determination of
the actual plant WUE and thus a simpler interpretation of plant processes and their relation to LAI. In general, WUE at our
study site varied between 0 and 5 g C (kg $H_2O)^{-1}$. This is consistent with the WUEs observed for northern grasslands (0–7 g

C (kg $H_2O)^{-1}$) (Tang et al., 2014).

The different management practices, such as fertilisation and the choice of grass cutting height, were slightly different in
the first and second year, which probably had an impact on the carbon balances. In June 2018, a conventional cutting height
of 6 cm was used, whereas in the other harvests the grass was cut at 15 cm. The higher cutting height may have enhanced the
regrowth of the grasses, especially in the more favourable weather conditions in 2019, and with a larger leaf area higher $CO_2$

uptake was observed right after the harvest. Only after the 6-cm harvest, the field turned to a net source of $CO_2$. With a low
cutting height, it was more likely that the grass was cut below the growing point, particularly in dry conditions, which affects
the stand longevity and stress tolerance (Jones and Tracy, 2018). As the weather was warm and dry during the harvest events
in June in both years, a higher cutting height may have served as a vital management improvement.

The field was mainly fertilised with organic substances, and thus carbon was imported to the system, affecting the net carbon

balance. After each of the fertilisation events with organic material, the respiration of the field seemed to increase, whereas
mineral fertilisation was not observed to have an immediate effect on $CO_2$ fluxes. Increased respiration was likely to occur due
to microbial activity of the organic fertilisers. Gilmanov et al. (2007) observed on a Danish agricultural grassland that, although
the application of manure increased respiration, also the plant uptake of $CO_2$ was notably higher than at the other sites studied.
Fornara et al. (2016) also concluded, based on their 43-yr study, that manure fertilisation substantially increased soil carbon

sequestration of a grassland ecosystem in Northern Ireland. Although the type of the organic fertiliser possibly plays a crucial
role, the application of carbon to the system has a direct effect on the carbon balance, but there is also an indirect effect on its
components $R_{eco}$ and GPP via soil and plant functions.

Concerning our final research question on the relation of possible carbon sink to the international "4 per 1000" carbon
sequestration initiative (Minasny et al., 2017), our results show that, on average, the field acted as a net annual carbon sink by

increasing the soil carbon content by 0.5% annually over the studied period. Thus the site fulfilled the goal of the "4 per 1000"
initiative and contributed to the short-term climate change mitigation. Furthermore, the annual carbon balance of our second
study year (–118 g C m$^{-2}$ yr$^{-1}$) is in the upper range of annual carbon sequestration potential (80–120 g C m$^{-2}$ yr$^{-1}$) that is
evaluated to be attainable with improved management practices (Lal, 2016). Thus, this study demonstrates the potential for a
positive impact of northern agricultural grasslands in terms of climate change mitigation.





## 4.2 Errors and uncertainties

Uncertainties with the data are mainly related to the gaps in the measurement data, which required gap-filling of those periods with modelled data. The length of a gap increases the related uncertainty, but in our data there were only three longer gaps (4, 8 and 9 days), which all occurred during the first winter, when temperatures were low and only minor fluxes could have been observed. All the other gaps were shorter than 3 days. However, each gap contributed to the uncertainty and were included in the carbon balance calculations. Further uncertainties, which were not included in the error estimates, were caused by the the soil temperature modelling for the first study year and the management flux estimates.

Carbon balance was calculated based on the ecosystem-atmosphere $CO_2$ fluxes and the inputs and outputs of harvest and fertilisation. Thus, no other gaseous carbon compounds, such as methane, were considered. Regina et al. (2007) reported that the annual methane exchange of a Finnish clay soil varied between –0.009 and 0.034 g $CH_4$ m$^{-2}$ yr$^{-1}$ during two years in 2000–2002. Thus, based on this estimate, the possible carbon emission from methane accounts for less than 1% of our annual carbon balance.

Leaching of dissolved carbon and emissions of volatile organic compounds may have had an effect on the annual carbon balance. Leaching of carbon from the agricultural soils is mainly driven by the meteorological and hydrological conditions (Manninen et al., 2018), but it is also affected by soil properties (Don and Schulze, 2008). Large variations in soil moisture and temperature and precipitation may increase the solubility of SOM. Generally, however, clay soils retain carbon better than other soil types. Furthermore, ploughing increases leaching as mineralisation of SOM is enhanced. Depending on precipitation and hydrological and chemical properties of the soil, carbon leaching on grasslands may equal approximately to 25% of the annual carbon balance calculated based on NEE, harvest and fertilisation (Kindler et al., 2011). At our study site, the effect of leaching on carbon balance could be assumed to be fairly small in both summers due to low soil moisture and low precipitation. On the other hand, during wet periods, the leaching may have had a small effect on carbon balance. However, a more precise carbon balance estimate would require further measurements, including leaching and other carbon-containing gases.

## 5 Conclusions

The agricultural grassland site located at Qvidja in southern Finland acted as a net carbon sink during the two years studied. The carbon balance of the first study year was –50 ± 68 g C m$^{-2}$ yr$^{-1}$ and in the second year it was –118 ± 24 g C m$^{-2}$ yr$^{-1}$. We estimated that on average the grassland exceeded the goal of the "4 per 1000" initiative intending to increase soil carbon content. The data presented here act as a basis for the future studies at this site that focus on the conversion from intensive agricultural practices towards more sustainable agricultural management and its impacts on the GHG fluxes on mineral soils in northern conditions. Further research with longer-term measurements would be needed to evaluate the persistence of carbon sequestration and storage. Longer time series would be also essential to study more closely the causes of the interannual variation of GHG fluxes and carbon and water balances at this site.



*Data availability.* The flux and meteorological data as well as the SOC measurements and LAI data are available at Zenodo
(https://doi.org/10.5281/zenodo.4297297, Heimsch et al. 2020).

**Appendix A: Effective phytomass index**

The PI was used to refine the gap-filling of GPP, especially in the case of long gaps in the nighttime data, based on which
GPP was parameterised. PI reflects the development of LAI but, being derived from the daytime NEE measurements, is more
dynamic than LAI and thus describes more precisely the course of the photosynthetic activity of plants (Aurela et al., 2001).
PI was derived from the net ecosystem $CO_2$ exchange data by selecting fluxes at high PAR levels. The PAR limit was set to
700 $\mu$mol m$^{-2}$ s$^{-1}$ from March to September and 200 $\mu$mol m$^{-2}$ s$^{-1}$ from October to February. The assumed respiration,
i.e. fluxes when PAR < 20 $\mu$mol m$^{-2}$ s$^{-1}$, was subtracted from from the NEE data. This was followed by averaging NEE and
$R_{eco}$ within a moving window, which was set to 3 days and increased to 5 or 7 days if necessary. Averaging was limited to the
harvest dates by decreasing the window size step-by-step to 1.5 days, and similarly increasing it after the harvest. An average
GPP was then calculated by subtracting $R_{eco}$ from NEE and normalising to unity to obtain PI. Daily PI values were used for
calculating the GPP fluxes. Due to the scarcity of respiration data in July in both years and of the daytime data in winter, linear
interpolation was applied to cover the missing daily PI values.

**Appendix B: Gap-filling of $CO_2$ fluxes**

The flux data set was separated into sections at the harvest dates, and gap-filling was done separately for these sections by first
calculating $R_{eco}$ and then GPP. The parameter $R_0$ was determined for each day from the nighttime data (PAR < 20 $\mu$mol m$^{-2}$
s$^{-1}$) with a 7-day moving window. If there were less than 24 measurements within the time window, its length was increased
by 1 day both in the beginning and at the end until enough data were obtained. $R_0$ was allowed to vary between 0.001 and 1
mg m$^{-2}$ s$^{-1}$. Similarly, the same minimum number of observations and a 3-day moving window was used for determining $\alpha$
and $GP_{max}$ from the observed NEE from which the estimated $R_{eco}$ had been subtracted. $\alpha$ and $GP_{max}$ were allowed to vary
between –0.1 and 0 mg $\mu$mol$^{-1}$, and –5.0 and 0 mg m$^{-2}$ s$^{-1}$, respectively. From 5 December 2018 to 26 March 2019 and
from 26 November 2019 to 15 March 2020, with no significant $CO_2$ uptake, a 5-day moving average was used to fill the gaps
in NEE.

**Appendix C: Gap-filling of energy fluxes**

The gaps in the net radiation ($R_n$) time series were filled with the monthly mean diurnal cycles. Soil heat flux (G) was not
measured at our site, so it was estimated from the energy balance closure during the periods when the other energy fluxes
were known. Gap-filling of G was done by assuming a constant ratio between G and $R_n$ (Liebethal and Foken, 2007). The
ratio of 0.24 was calculated with linear regression from the daytime data (between 10:00–15:00). The sensible and latent heat
fluxes ($Q_H$ and $Q_E$, respectively) were gap-filled based on the procedure described by Kowalski et al. (2003). The gaps in the

daytime $Q_H$ ($R_n$>0) were filled with monthly linear regression with $R_n$. The nighttime gaps in $Q_H$ ($R_n$<0) were filled with the corresponding $R_n$ values. The gaps in the daytime $Q_E$ were filled in such a way that the monthly mean energy balance closure was achieved. The nighttime gaps in $Q_E$ were set to 0.

## Appendix D: Management effect on fluxes

The immediate effect of management on the measured NEE and WUE were investigated by comparing the mean values of five days before and after the management day (Table A1).

*Author contributions.* JL and TL planned the flux measurements and TL was responsible for the setup. JPT made the post-processing data corrections and calculated the flux footprint. HV and MK developed the gap-filling code. LH filtered the data and carried out the data analysis. JH provided the soil carbon data and ON processed the Sentinel-2 LAI data. LH, AL, JPT and LK prepared the manuscript with contributions

from all co-authors.

*Competing interests.* The authors declare that they have no conflict of interest.

*Acknowledgements.* This study was supported by SITRA, Business Finland [grant 6905/31/2018] and The Strategic Research Council at the Academy of Finland [grant no 327214]. We acknowledge MSc student Niina Ruoho for skillful technical assistance. Qvidja farm owners and staff, especially Pekka Heikkinen and Jonathan Nylund are greatly acknowledged for diverse practical assistance and management of the

field.





**Table A1.** Mean flux and meteorological conditions 5 days before and after management. The management day is not included.

| | NEE (mg $CO_2$ m$^{-2}$ s$^{-1}$) | | WUE (g C kg$^{-1}$ H$_2$O) | | PAR ($\mu$mol m$^{-2}$ s$^{-1}$) | | Air T (°C) | | Precipitation (mm) | | Soil moisture (m$^3$ m$^{-3}$) | |
|---|---|---|---|---|---|---|---|---|---|---|---|---|
| | Before | After | Before | After | Before | After | Before | After | Before | After | Before | After |
| Harvest 12 Jun 2018 | −0.28 | 0.03 | 2.6 | 0.8 | 563 | 646 | 12.5 | 16.4 | 0 | 0 | 0.24 | 0.23 |
| Fertilisation 16 Jul 2018 | −0.27 | 0 | 1.9 | 2.0 | 516 | 480 | 19.9 | 22.4 | 0 | 8.3 | 0.23 | 0.21 |
| Cutting 21 Aug 2018 | na | −0.02 | na | 1.8 | na | 290 | na | 17.2 | 0 | 1.2 | na | 0.23 |
| Fertilisation 24 Aug 2018 | −0.10 | −0.02 | 2.0 | 1.3 | 382 | 273 | 15.7 | 14.5 | 6.7 | 10.4 | 0.24 | 0.24 |
| Harvest 23 Sep 2018 | −0.10 | −0.02 | 2.4 | 1.3 | 183 | 226 | 15.1 | 8.6 | 0.7 | 8.6 | 0.36 | 0.34 |
| Fertilisation 8 May 2019 | −0.17 | 0.17 | 2.4 | 1.8 | 367 | 324 | 3.2 | 9.6 | 1.5 | 3.7 | 0.37 | 0.29 |
| Harvest 11 Jun 2019 | −0.50 | −0.03 | 2.9 | 2.2 | 627 | 412 | 21.2 | 15.8 | 0.4 | 0.7 | 0.23 | 0.20 |
| Fertilisation 20 Jun 2019 | −0.08 | −0.08 | 1.8 | 1.8 | 601 | 622 | 17.5 | 17.0 | 0 | 2.5 | 0.20 | 0.20 |
| Harvest 20 Aug 2019 | −0.25 | −0.02 | 3.1 | 1.4 | 268 | 354 | 16.0 | 15.9 | 12.6 | 9.9 | 0.35 | 0.36 |



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
