# Peer review of "Carbon dioxide fluxes and carbon balance of an agricultural grassland in southern Finland"

_Biogeosciences, 2020_

## Referee Comment (RC1) · Anonymous Referee #1 · 4 Jan 2021

Heimsch et al. present data of an agriculturally used grassland in southern Finland that recently transitioned from intensive to sustainable management. Although the results are interesting, the large differences between the years (like the number and heights of harvests/cuts, the type of fertilization, the amount of precipitation, the progression since seeding and the reseeding of a different species composition) hinder the authors to draw specific conclusions as to what the changes are related to. In this regard, I am not sure if comparing the years makes sense.

The authors also state, that their soil temperature sensor malfunctioned for the first year and that the soil temperature had to be modelled for the whole year. This modelled soil temperature was then used to model the ecosystem respiration, which was used to gap fill the eddy covariance data for the carbon balance. In a publication focusing on the carbon balance the major variables needed should only be gap filled over a short period, not a whole year (50% of the duration of this study). Beside these two major problems, when calculating the ecosystem respiration, the authors use a fixed value for the sensitivity of the ecosystem respiration and state that it describes the temperature response of the soil respiration. In a highly dynamic grassland, the changes in the respiration of the above ground biomass should not be missed. Thus, sensitivity parameter should be based nighttime NEE data using a moving time window to account for these changes. I recommend rejecting this publication and let the authors recalculate the data and rewrite the manuscript with a different angle as the problems mentioned will likely not be solved in one major revision and result in a different publication. As a possible solution for their soil temperature modelling problem, I suggest that the authors try to use the air temperature to calculate the ecosystem respiration for both years as many other studies do. (see https://doi.org/10.5194/bg-9-5243-2012 for consequences)

---

## Referee Comment (RC2) · Anonymous Referee #2 · 29 Jan 2021

The ms introduces a new data set on eddy fluxes of $CO_2$ and $H_2O$ observed above a managed grassland in southern Finland. Given the huge number of eddy covariance $CO_2$ studies, and especially of multi-year and multi-site analyses based on networks and infrastructures such as Fluxnet, ICOS, NEON and others, one could ask whether it is relevant to publish an additional study covering only one site over two years. However, apart from the fact that this ms is very well and thoroughly written, it deals with an ecosystem type for which data are still scarce. The majority of eddy towers is located in natural ecosystems, whereas agricultural sites in general are underrepresented. It is necessary to focus more on these sites because the driving factors of soil-vegetation-atmosphere exchange are much less well understood, and because the magnitude of the fluxes is strongly influenced by management activities. This makes the results from

such case studies highly relevant not only for basic scientific analyses and for monitoring adaptation to climate change, but also for developing mitigation concepts. The data seem to be of high quality, and most of the conclusions are correctly drawn. It was a pleasure to read the ms, finally yet importantly because it is written in such a way as to make the study fully reproducible, which is clearly an advantage, from the readers' perspective, in comparison to papers that refer to a chain of other publications when describing the methods. Therefore, I recommend that the ms be published in Biogeosciences, provided the authors submit a moderately revised version that considers the following specific points.

Lines 83-84: Clover is not a grass – please revise the sentence.

Line 109: Was only the zero point of the $CO_2$ channel of the LI-7200 calibrated? No calibration of the span and/or the $H_2O$ measurements? This is the only point in the Methods section that worries me a little bit, because from my experience I would say that this type of analyser requires a complete calibration about once a year.

Line 144: I had expected that the data coverage would have been described in more detail here: Was it spread evenly across seasons and/or times of the day? The temporal distribution of the gaps can have a considerable effect on the reliability of the gap filling and potential biases. Later I realised some comments on this in the appendix, but I would recommend adding a sentence or two about this in the main text.

Line 266: Regarding the relation to 'mean soil moisture' it is important which soil depth is referred to. Please add this information to the ms text.

Line 322 ff: I am not really sure about the definitions of the expressions of $CO_2$ exchange used here and in the subsequent paragraphs (this comment refers also to the 'uptake' mentioned in lines 338 and 340 – is this NEE or GPP or something else?). How is for instance the 'total carbon balance' defined, when for example comparing it to the nomenclature used by Kutsch et al. (2010) in AGEE 139, 336-345: Is it net biome production NBP? Or full field balance or farm gate balance? Please clarify.

Line 345: Strictly speaking, the enhanced root system wouldn't directly enhance carbon uptake, but enhance water availability and avoid drought stress, thereby indirectly enable a higher GPP (but also enhance respiration?!). Please explain this effect more accurately.

Line 353: I think the reference to Pirasteh et al. is a very odd choice given that this effect has been well known and understood for many decades.

Lines 361-363: With respect to the WUE analysis, I strongly recommend an additional approach to disentangle transpiration from total ET in order to make the statements on WUE more meaningful. I understand that the ET components were not measured separately, but assuming that soil evaporation can be neglected in a grassland with complete canopy cover, the data set could for example be split into rainy and rainless days, and the rainless days be analysed separately. This would enable an alternative calculation of WUE from GPP and T if it is further assumed that, on rainless days, interception evaporation can be neglected, too.

Line 395: The last "the" has to be deleted.

Line 396: See above: Which processes or flux components are exactly included in the expression "management flux"? I think the answer to research question no. 3 on carbon sequestration and offsetting carbon emissions depends strongly on the definition of the boundary of the system under consideration (ecosystem – field – farm gate?).

---

## Author Comment (AC1) · 19 Feb 2021

We would like to thank the reviewer for the constructive comments. They allowed us to substantially improve the manuscript. We have taken them all into account and made accordingly major revisions to the MS. Please find below the detailed responses on how we have addressed the concerns.

*"The authors also state, that their soil temperature sensor malfunctioned for the first year and that the soil temperature had to be modelled for the whole year. This modelled soil temperature was then used to model the ecosystem respiration, which was used to gap fill the eddy covariance data for the carbon balance. In a publication focusing on the carbon balance the major variables needed should only be gap filled over a short period, not a whole year (50% of the duration of this study)."*
*"As a possible solution for their soil temperature modelling problem, I suggest that the authors try to use the air temperature to calculate the ecosystem respiration for both years as many other studies do."*

> We agree with the reviewer that this was not an optimal solution even though we modelled the soil temperature using air temperature and showed that the model performance was good. Therefore, instead of using the soil temperature model, we changed the temperature used in gap-filling model from soil temperature to air temperature as the reviewer suggested. This had only a minor effect on the results.

*"the authors use a fixed value for the sensitivity of the ecosystem respiration and state that it describes the temperature response of the soil respiration. In a highly dynamic grassland, the changes in the respiration of the above ground biomass should not be missed. Thus, sensitivity parameter should be based nighttime NEE data using a moving time window to account for these changes."*

> This is a very valid point. We have adjusted our model so that the $E_0$ parameter is determined within the same moving window as $R_0$.

> While revising the manuscript, we performed gap-filling several times with varying settings and ended up having the best performance by fitting $E_0$ to the data with the same window as $R_0$. Furthermore, we decided to remove the effective phytomass index (PI) from the light response fitting to increase the accuracy of the fits during the wintertime. Due to the occasional negative fluxes and fairly high positive fluxes, the gap-filling of wintertime data was more accurate without PI. In addition to these changes, we re-considered and improved the uncertainty analysis. As a result of this change, the field proved to be a carbon sink in both years. We modified the method description and the discussion of results accordingly.

*"Although the results are interesting, the large differences between the years (like the number and heights of harvests/cuts, the type of fertilization, the amount of precipitation, the progression since seeding and the reseeding of a different species composition) hinder the authors to draw specific conclusions as to what the changes are related to. In this regard, I am not sure if comparing the years makes sense."*
*"I recommend rejecting this publication and let the authors recalculate the data and rewrite the manuscript with a different angle as the problems mentioned will likely not be solved in one major revision and result in a different publication."*

The referee is correct with this statement that the data only cover two years and that this hinders us from properly evaluating the interannual variation. We agree that such analysis would definitely need more data and possibly some help from modelling to interpret the effect of different factors contributing to the variation. Thus, we have modified the manuscript so as not to highlight the differences between the years. In practice, we removed the research question about the year-to-year differences. Instead, we raised a question addressing the characteristics of the $CO_2$ exchange dynamics and the overall annual carbon balance. As a consequence of this angle change, the discussion was adjusted in the revised MS.

Although, at this stage, we are not able to comprehensively study the interannual differences and drivers, this study is important and very timely as this kind of agricultural studies are scarce in the boreal region. Furthermore, there is an urgent need to quantify the GHG fluxes and budgets of agriculture in order to further develop sustainable management actions to decrease the various negative environmental impacts of current agricultural practices. It is not self-evident that for example the so-called climate-smart practices, which are proven to sequester carbon in temperate regions, are as beneficial in the boreal region, where the climate as well as the cultivated species and varieties are different. Thus, the present analysis can serve as a valuable baseline for further studies and more detailed analyses on the interannual changes and effects of improved farming practices.

---

## Author Comment (AC2) · 19 Feb 2021

We would like to thank the reviewer for the helpful comments, which allowed us to improve our manuscript. We present below the detailed responses on how we have addressed the comments.

*"Lines 83-84: Clover is not a grass – please revise the sentence."*

Yes, we revised this sentence.

*"Line 109: Was only the zero point of the CO2 channel of the LI-7200 calibrated? No calibration of the span and/or the H2O measurements? This is the only point in the Methods section that worries me a little bit, because from my experience I would say that this type of analyser requires a complete calibration about once a year."*

Yes, we agree and clarified the calibration procedure in the revised MS. The gas concentrations were regularly checked by measuring the $CO_2$ zero and span gases and comparing the $H_2O$ data to molar ratio calculated from Humicap data. $CO_2$ concentration was calibrated with zero and span gases when needed. There was no need to calibrate $H_2O$ gas during the measurement period.

*"Line 144: I had expected that the data coverage would have been described in more detail here: Was it spread evenly across seasons and/or times of the day? The temporal distribution of the gaps can have a considerable effect on the reliability of the gap filling and potential biases. Later I realised some comments on this in the appendix, but I would recommend adding a sentence or two about this in the main text."*

We added a few sentences about the distribution of gaps in winter- vs. summertime and day- vs. nighttime to the revised MS. Nighttime data coverage for the whole $CO_2$ data set was 33% and for the $H_2O$ data 11%. Daytime data coverage for $CO_2$ was 55% and for $H_2O$ 49%. Summertime (from April to September) and wintertime (from October to March) data coverage for $CO_2$ were 48% and 38%, and for $H_2O$ 41% and 16%, respectively.

*"Line 266: Regarding the relation to 'mean soil moisture' it is important which soil depth is referred to. Please add this information to the ms text."*

Yes, we added the information about the depth of soil moisture measurements.

*"Line 322 ff: I am not really sure about the definitions of the expressions of CO2 exchange used here and in the subsequent paragraphs (this comment refers also to the 'uptake' mentioned in lines 338 and 340 – is this NEE or GPP or something else?). How is for instance the 'total carbon balance' defined, when for example comparing it to the nomenclature used by Kutsch et al. (2010) in AGEE 139, 336-345: Is it net biome production NBP? Or full field balance or farm gate balance? Please clarify."*

Indeed, the carbon balance represents NBP as in Kutch et al. (2010). We clarified this in the revised Section *2.5 Carbon balance* to which we added a definition of the system boundaries adopted in our study. We defined the study system as the agricultural field ecosystem. Concerning the $CO_2$

uptake term on lines 338–340, we emphasized that the net uptake of atmospheric $CO_2$ is meant here.

*"Line 345: Strictly speaking, the enhanced root system wouldn't directly enhance carbon uptake, but enhance water availability and avoid drought stress, thereby indirectly enable a higher GPP (but also enhance respiration?!). Please explain this effect more accurately."*

We added a more accurate description that the effect is indeed indirect.

*"Line 353: I think the reference to Pirasteh et al. is a very odd choice given that this effect has been well known and understood for many decades."*

We updated the reference here to Loftfeld (1921).

*"Lines 361-363: With respect to the WUE analysis, I strongly recommend an additional approach to disentangle transpiration from total ET in order to make the statements on WUE more meaningful. I understand that the ET components were not measured separately, but assuming that soil evaporation can be neglected in a grassland with complete canopy cover, the data set could for example be split into rainy and rainless days, and the rainless days be analysed separately. This would enable an alternative calculation of WUE from GPP and T if it is further assumed that, on rainless days, interception evaporation can be neglected, too."*

We agree. We found that most of the rainy days were already eliminated from the analysis as we used only the days when the latent heat flux was greater than 30 W m$^{-2}$. However, we further modified the filtering and removed all the rainy days. As a result of this change, 10 more data points were eliminated, and the results were affected very little. The relationship between WUE and LAI (Fig. 5) was not affected. Below updated Figure 6.

[Figure]

Figure 6. Daily water use efficiency (WUE) during two growing seasons.

*"Line 395: The last "the" has to be deleted."*

Yes, that was deleted.

*"Line 396: See above: Which processes or flux components are exactly included in the expression "management flux"? I think the answer to research question no. 3 on carbon sequestration and offsetting carbon emissions depends strongly on the definition of the boundary of the system under consideration (ecosystem – field – farm gate?)."*

Management flux means the sum of positive outflow as yield and negative inflow as organic fertiliser. The term was clarified in the revised text (this line and earlier). Concerning the carbon sequestration of the field, it is true that its estimation depends greatly on the definition of the system boundary. We clarified the system boundary in Section *2.5*, as described above, by defining the study system as the agricultural field ecosystem. We also added a sentence to the discussion to highlight that the carbon balance is affected strongly by the carbon fluxes caused by fertilisation and harvests.

---

## Author Response (AR2)

We want to thank the editor for his time and efforts! The raised comments were worth noticing, and we edited the MS accordingly. The most significant changes considered the latter part of the MS to which we added discussion on the limitations of the selected methodology. Please find our point by point answers below.

P1L10-11: The sentence was removed.

P2L29-30: Clarification was added to the sentence that the 0.4% means the average increase in soil carbon content on all soils. Furthermore, we added a sentence to highlight that the increase of soil carbon is related to soil properties, e.g. clay content.

P2L44-45: Evapotranspiration corrected to transpiration.

P2L54: "better" changed to "improved".

P2L57: "global monitoring and verification system" changed to "global measuring, reporting and verification systems".

P3L62-63: We removed the reference to carbon sequestration from the abstract.

P3L64: "Nordic countries" changed to "Northern European countries".

P3L88: "Repair seeding" corrected to "oversowing".

P4L93: The cutting height was increased by the farmers because they want to apply practices that are presumed to increase carbon sequestration.

P4L95: "was" changed to "is".

P4L101: "was" changed to "were".

P4L102: "was" changed to "were".

P6L151: Data coverage was calculated for all 30-minute periods within the two study years, i.e. 35 088 time points (including the leap day 29.2.2020). We added this information to the revised MS.

P7L172: "net exchange" corrected to "net ecosystem exchange".

P8L195: It is mentioned after the balance equation that other carbon losses, such as leaching, are not taken into account in the balance. We included discussion on the possible importance of these components in the revised MS (see later).

P9L208: Yes, this part of the sentence was removed.

P9L225: We added a note that the signal is dominated by transpiration.

P9L226: "content" changed to "storage".

P10L242: In this case, the mean air temperature was calculated for the main growing season, i.e. May to September. It was clarified by changing "these months" to "these periods" referring to the previous sentence.

P10L246-249: The wintertime fluxes were very small, even compared to the summertime nocturnal fluxes. In March 2019, the field seemed to uptake $CO_2$ during a few days when the snow cover was thin and light availability was sufficient. Other than that, no notable $CO_2$ uptake was observed outside the thermal growing season. As the $CO_2$ fluxes during those days played an insignificant role in the whole carbon balance and the snow depth measurements were taken outside the footprint area, we prefer not to discuss this phenomenon in the manuscript.

P10L256: Subtitle changed to "$CO_2$ and $H_2O$ fluxes".

P15L316-317: We clarified the relationship between the carbon balance and SOC.

P16 Figure 7 caption: Obviously, a greater sample size would be desirable, but taking and analysing 1-m core samples is a resource-intensive task. However, these five samples were taken so as to obtain representative data within the main footprint area of the eddy covariance measurements. We added a few sentences to the discussion to address the uncertainties and limitations related to the fairly small sample size.

P16-17L337-339: Based on the comment, we removed the comparison of our carbon balance to the $CO_2$-eq. balances of European agricultural grassland sites

P17L356: "activities" added.

P17L356: "was" changed to "were".

P19L416-423: The paragraph was reformulated, and limitations of the current research, such as a missing leaching estimate and the small number of soil samples, were mentioned.

P20L442-447: This part of the text was rephrased, and the discussion of limitations was enhanced. Furthermore, the need for further research was highlighted.

P20L451: Yes, we agree. The sentence was rewritten, and the conclusion was modified indicating that the field has a potential to mitigate climate change but further studies are needed to more accurately quantify the long-term carbon balance and to address the contribution of other carbon fluxes, such as leaching and other carbon containing gases.